# Polyphenol Supplementation Reverses Age-Related Changes in Microglial Signaling Cascades

**DOI:** 10.3390/ijms22126373

**Published:** 2021-06-14

**Authors:** Ahmad Jalloh, Antwoine Flowers, Charles Hudson, Dale Chaput, Jennifer Guergues, Stanley M. Stevens, Paula C. Bickford

**Affiliations:** 1Center of Excellence for Aging and Brain Repair, Departments of Neurosurgery and Brain Repair, and Molecular Pharmacology and Physiology, USF Morsani College of Medicine, 12901 Bruce B. Downs Blvd, MDC 78, Tampa, FL 33612, USA; ahmadjalloh@usf.edu (A.J.); aflow86@gmail.com (A.F.); 2Research Service, James A Haley VA Hospital, Tampa, FL 33620, USA; chudson1@usf.edu; 3Department of Cell Biology, Microbiology and Molecular Biology, University of South Florida, Tampa, FL 33620, USA; chaput@usf.edu (D.C.); j.guergues@vanderbilt.edu (J.G.); smstevens@usf.edu (S.M.S.J.)

**Keywords:** microglia, aging, polyphenols, mass spectrometry, bioinformatics

## Abstract

Microglial activity in the aging neuroimmune system is a central player in aging-related dysfunction. Aging alters microglial function via shifts in protein signaling cascades. These shifts can propagate neurodegenerative pathology. Therapeutics require a multifaceted approach to understand and address the stochastic nature of this process. Polyphenols offer one such means of rectifying age-related decline. Our group used mass spectrometry (MS) analysis to explicate the complex nature of these aging microglial pathways. In our first experiment, we compared primary microglia isolated from young and aged rats and identified 197 significantly differentially expressed proteins between these groups. Then, we performed bioinformatic analysis to explore differences in canonical signaling cascades related to microglial homeostasis and function with age. In a second experiment, we investigated changes to these pathways in aged animals after 30-day dietary supplementation with NT-020, which is a blend of polyphenols. We identified 144 differentially expressed proteins between the NT-020 group and the control diet group via MS analysis. Bioinformatic analysis predicted an NT-020 driven reversal in the upregulation of age-related canonical pathways that control inflammation, cellular metabolism, and proteostasis. Our results highlight salient aspects of microglial aging at the level of protein interactions and demonstrate a potential role of polyphenols as therapeutics for age-associated dysfunction.

## 1. Introduction

Aging progressively compromises an organism’s ability to maintain homeostasis, increases its susceptibility to disease and death, and is the major risk factor for most neurodegenerative diseases [1]. Alzheimer’s, Parkinson’s, and similar disorders have a complex molecular origin with multiple etiologies that underscore their debilitative symptoms; however, the immune system is a common factor among progression of these diseases [2]. The central nervous system (CNS) mediates immunity through the neuroimmune system, which is a set of cells that maintain homeostasis by regulating neuroinflammation. Microglia are among the core effector cells of this system. These myeloid-derived cells are the brain’s resident macrophage that acquire different functions by dynamically altering their morphology [3]. Microglia form contacts with neurons and monitor the brain parenchyma for pathogenic invasion or cellular damage [4]. Quiescent microglia possess a highly ramified structure that interacts with neurons [5], and samples their surroundings. Microglia enact CNS immunity through the initiation and resolution of the neuroimmune response upon binding cellular debris via puringergic receptors or pathogens via Toll-like receptors (TLRs) [6]. Microglia will polarize toward a phenotype that promotes antigen presentation during this process [7]. Pro-inflammatory microglia secrete molecules such as tumor necrosis factor α (TNFα), interleukin-1β (IL-1β), and interleukin-6 (IL-6), and they generate reactive oxygen species (ROS) and reactive nitrogen species (RNS) to limit damage to CNS tissues [8]. Then, microglia are polarized by anti-inflammatory molecules such as interleukin-4 (IL-4) and interleukin-10 (IL-10) toward a phenotype that supports wound healing and debris clearance via phagocytosis in a secondary process that resolves the immune response [9]. These processes are highly regulated and essential for CNS homeostasis; however, aging can make microglia and their activation states become dysfunctional and dysregulated [10].

Previous studies have described a “priming” effect in aged microglia. Primed microglia are hyper-responsive to pro-inflammatory stimuli and show a blunted response to anti-inflammatory signals [11]. This phenomenon increases neuronal vulnerability and facilitates microglia-induced neurotoxicity that plays a demonstrated role in Alzheimer’s [12], ALS [13], and Parkinson’s [14]. Aging, as such, modulates inflammatory signaling cascades toward pathology. Microglia from aged mice possess higher baseline cytokine expression compared to microglia from young mice; furthermore, aged microglia show higher pro-inflammatory gene expression after TNFα stimulation and lower anti-inflammatory gene expression after IL-4 stimulation [15]. Other research confirms that aged microglia overexpress inflammatory genes such as major histocompatibility complex II (MHCII) and CD68 that regulate antigen presentation [16], integrins such as CD11b [17] and cytokines such as TNF, IL-1β, and IL-6 [18]. Microglial activation and tau deposition increases in WT and 3xTg AD transgenic mice after acute intracranial or systemic LPS stimulus [19,20]. When considered together, these studies emphasize the molecular consequences on the CNS brought on by microglial aging.

Longevity and well-being depend on limiting CNS damage from neuroinflammation. Polyphenol-rich diets are one approach to mitigating the deleterious effects of prolonged neuroinflammation and reducing the risk of neurodegenerative disease. Research over the last decade has established a solid body of evidence for the clinical effectiveness of polyphenols on age-related decline [21]. One study demonstrated that dietary (−)-epigallocatechin-3-gallate (EGCG), which is a catechin found in green tea, ameliorated skeletal muscle insulin resistance and reduced liver fat accumulation in senescence-accelerated mice [22]. This process occurred inhibition of the PI3K/Akt pathway, decreased glucose transporter type 4 (GLUT4) expression, and a downregulated mechanistic target of rapamycin (mTOR) activity via decreased sterol regulatory element-binding proteins-1c (SREBP-1c) and nuclear factor kappa-light-chain-enhancer of activated B cells (NF-κB) expression. Another study showed that quercetin, a flavonoid found in blueberries, improved motor coordination and spatial learning in aged rats concomitant with reduced hippocampal NF-κB and elevated sirtuin 1 expression [23]. Molecules such as vitamin D and L-carnosine have proven effectiveness in treating symptoms of aging. Vitamin D reduced microglial pro-inflammatory cytokine and ROS production while inducing anti-inflammatory cytokine expression in a Parkinson’s disease animal model [24]. Likewise, L-carnosine significantly reduced amyloid-Beta (Aβ) deposition in the prefrontal cortex, hippocampus, and hypothalamus of aged rats, decreased serum corticosterone, and restored locomotor activity [25]. Polyphenols are well-characterized in aging studies and beneficial, tempering age-related decline.

In our present study, we used a mass spectrometry (MS)-based proteomics approach to compare microglia extracted from young (≈3–6 months) and aged rats (≈20 months). Our first experiment highlighted age-related changes in canonical microglial signaling pathways, annotated potential upstream transcriptional regulatory molecules involved in the aging phenotype, and predicted several changes in cell function related to microglial aging. Our second experiment elucidated the modulatory effect of the polyphenol supplement NT-020 (a proprietary blend of blueberry, green tea, vitamin D3, and l-carnosine). This formulation was chosen specifically based upon a study of hundreds of potential ingredients where top candidates were tested for the ability to promote stem cell proliferation in vitro, and this combination was found to be more than additive [26]. Follow-up studies demonstrated that the administration of this supplement to mice protects cells from oxidative insults during ischemic stroke [27]. Of relevance to this investigation, we have previously demonstrated that NT-020 improves neuronal progenitor cell proliferation while improving cognitive function and reducing brain inflammation in aged rats [28,29]. In this study, we found that 30-day dietary supplementation in aged rats reversed the predicted age-related trends we observed in canonical microglial protein networks, transcriptional regulatory patterns, and cell function pathways. These results explicate a systems-level understanding of molecular aging in rat microglia extrapolated at the protein level and provide further evidence for NT-020 as a potential therapeutic for age-related microglial dysfunction.

## 2. Results/Discussion

### 2.1. NT-020 Supplementation Alters Microglial Protein Expression in Aging Rats

Proteomic analysis identified 2563 proteins across all experimental groups (*YNG*, *YNG-NT-020*, *OLD*, *OLD-NT-020*). After filtering for quantifiable proteins, we detected 1154 proteins in the *YNG* and *OLD* comparison with 197 significantly differentially expressed (*p* < 0.05, Welch’s *t*-test) proteins identified as part of our *AGING* comparison (OLD/YNG). We identified 967 and 1358 quantifiable proteins in the *YNG-NT-020* (YNG-Tx/YNG) and *OLD-NT-020* comparisons (OLD-Tx/OLD), respectively. We found 36 proteins significantly differentially expressed in the *YNG-NT-020* comparison and 144 in the *OLD-NT-020* comparison. Further statistical filtering was also implemented using a z-score cutoff (|z-score| > 1) to provide a more stringently filtered list in terms of lower false discovery rate (FDR) (see Appendix A for full and filtered protein lists by comparison). However, we used only differentially expressed proteins filtered by Welch’s *t*-test for subsequent bioinformatics analyses. This filtering allowed a wider search space to facilitate the identification of significant as well as trending global-scale activity changes in pathways related to age and antioxidant (NT-020) treatment.

In the *AGING* and *OLD-NT-020* comparisons, we plotted the Welch’s *T*-test difference vs. −log_10_(*p*-value) for each protein to obtain the volcano plots shown in Figure 1A,B. Highlighted in red (upregulated) and green (downregulated) are those differentially expressed proteins that passed the z-score cutoff. Our analyses excluded the *YNG-NT-020* comparison because the low number of differentially expressed proteins did not generate any significant changes to canonical pathways nor cell functions (data not shown).

Here, we observed a shift in the expression of several immune regulators including integrin alpha M (*ITGAM*), also known as CD11b, Complement component 3 (*C3*), heat-shock protein 90 beta family member 1 (*HSP90B1*), and progesterone receptor membrane component 1 (*PGRMC1*) that were upregulated in aged microglia but downregulated in aged microglia from NT-020 supplement rats. We also detected upregulation greater than 1.5-fold with age and downregulation less than 1.5-fold with NT-020 treatment in other molecules with a demonstrated capacity to induce the microglial inflammatory. Among those molecules were hexokinase 2 (*HK2*) [30], interferon regulatory factor 5 (*IRF5*) [31], and Rac family small GTPase 1 (*RAC1*) [32]. We used the Wes instrument to validate proteins in our MS data by quantifying C3, HSP90B1, ITGAM, and PGRMC1. We confirmed the trends we observed with mass spectrometry in the chemiluminescent intensities for each protein normalized to their respective values for GAPDH (Figure 1C–F) even for proteins such as HSP90B1 and ITGAM that had *p*-values < 0.05 but z-scores < 1. Both HSP90B1 and ITGAM showed higher LFQ intensity values measured by mass spectrometry (compared to C3 and PGRMC1), which typically can be measured with lower variance, highlighting that subtle changes in differential protein expression can be accurately detected as confirmed by an orthogonal validation approach. We used a more relaxed significance cutoff (*p* < 0.10) in the canonical and functional pathway analyses we discuss below to incorporate a greater number of proteins and enable wider coverage of relevant pathways.

Our results corroborate other literature findings regarding immune cell response to polyphenol supplementation. These studies relate the specific compounds comprising NT-020 such as green tea-derived EGCG and quercetin found in berry extracts to immunomodulatory function. EGCG was shown to downregulate and bind CD11b (ITGAM) on B and CD8^+^ T cells and reduce adhesion and decreases migration at infiltration sites [33,34]. Likewise, quercetin supplementation was shown to inhibit dendritic cell maturation and impede disease progression in mouse models of atherosclerosis via decreased nuclear NF-κB translocation [35]. Those effects were also concomitant with decreased inflammatory marker expression and reduced ability to stimulate T cells in vitro. Another study determined vitamin D_3_ secreted from neurons attenuated LPS-induced microglial inflammation and deleting the vitamin D_3_ receptor in mice increased CNS autoimmunity [36]. Finally, l-carnosine was shown to reduce Aβ-induced pro-inflammatory microglial activation in a transforming growth factor beta 1 (TGF-β1)-mediated process [37]. Our results expand upon these studies on polyphenols and immune cell function by demonstrating anti-inflammatory effects in aging microglia.

### 2.2. Gene Ontology Analysis Highlights Global Functional Changes in Aging Tat Microglial Proteome with NT-020 Diet

We performed gene ontology on our MS results to visualize global changes in the microglial proteome with age and contrast those changes in aging rats supplemented with NT-020. We assessed enrichment in GO terms queried via *g:Profiler* [38] in the *AGING* (Figure 2A) and *OLD-NT-020* (Figure 2B) comparisons. The *YNG-NT-020* comparison is not shown in this, and any subsequent bioinformatics results as the low number of differentially expressed proteins produced no GO annotations during computation, nor is this sufficient for meaningful analysis in IPA. The data are reported in the supplemental data tables.

We observed enrichment for several microglia-relevant terms in our both comparisons, particularly cellular response to cytokine stimulus, integrin alpha-M beta 2 complex, antigen presentation and processing in AGING and synapse pruning, phagocytic vesicle, and innate immune system in OLD-NT-020. We observed a reversal in trends from AGING to OLD-NT-020 when we ranked the top 10 terms annotated in both comparisons by z-score as visualized by circle plot (Figure 2C,D). Our gene ontology analyses provide a cursory overview as to how microglia in an aging animal differ with polyphenol supplementation; namely, our data indicate a shift in metabolic processes in aging microglia that shifts in the opposite direction with polyphenol supplementation. We found that upregulated protein expression enriched GO terms in *AGING*, whereas downregulated protein expression enriched GO terms in *OLD-NT-020*.

### 2.3. Pathway Analysis Identifies Canonical Pathways Driving Age-Related Changes in Microglial Proteome Reversed with NT-020 Diet

We used bioinformatic analysis to explore changes in signaling networks in the microglial proteome in each of our comparisons. We calculated microglia-related canonical pathways through the core analysis function in IPA for the *AGING* comparison (Figure 3A) and contrasted their states in the *OLD-NT-020* comparison (Figure 3B). We do not show the *YNG-NT-020* comparison from canonical pathway analysis because low protein enrichment failed to manifest annotated pathways.

Appendix A lists the full set of canonical pathways and abbreviated terms belonging to each pathway. The −log_10_(*p*-value) was derived from Fisher’s exact test (right-tailed), and the z-score in this context describes a predicted activation state inferred from dataset protein expression. Below, we focus our discussion of these canonical pathways and the changes accompanying NT-020 supplementation in the three pathways with the largest change in activation z-score across the two comparisons.

Again, we observed a similar pattern in canonical pathways as in our differential expression and GO analyses: each pathway activated in *AGING* had the opposite regulation in *OLD-NT-020*. *Unfolded Protein Response Signaling* (*UPR*) was the most activated pathway predicted in *AGING* (*z* = 2.828) and is a highly conserved cellular mechanism that is integral for maintaining homeostasis [39]. Neurodegenerative disorders such as Alzheimer’s [40] and Parkinson’s [41] have pathology that features aberrant UPR-related and by proxy, Endoplasmic Reticulum Stress (ER Stress)-related signaling. Our finding in *AGING* underscores emerging literature describing the role of proteostasis and its dysregulation in a neurodegenerative context [42,43] but expands this context by also implicating aging microglia. Abnormal or overactive UPR signaling is a likely culprit or by-product of the aging microglial phenotype but requires further study to elucidate properly. Furthermore, our data showed that NT-020 supplementation garnered an inhibitory effect in aging microglia. UPR signaling was predicted to be inhibited (*z* = −2.236) in the *OLD-NT-020* comparison. This prediction is significant because the exploration of polyphenolic compounds and their influence on age-related UPR and ER stress signaling is an increasing area of interest [44]. A recent study showed in an Alzheimer’s mouse model of obesity that a single oral dose of EGCG significantly decreased hippocampal UPR activation via decreased phosphorylated-EIF2α (S51), activation factor 4 (ATF4), and CATT-enhancer-binding protein homologous protein (CHOP) [45]. Reversing UPR dysfunction in aging microglia is crucial to re-establishing cellular homeostasis, and our data suggest a role for dietary polyphenols in this process.

*Interleukin-8* (*IL-8*) *signaling* was another pathway that demonstrated reversal from *AGING* to *OLD-NT-020*. IL-8 is a pro-inflammatory macrophage chemokine that transduces signals from lipopolysaccharide and other cytokines [46]; moreover, its inhibition attenuates microglial activation in murine Alzheimer’s models [47]. We observed predicted activation in this pathway in our *AGING* comparison (*z* = 2.000) that reversed to inhibition in our *OLD-NT-020* comparison (*z* = −0.816). This result suggests a possible interaction between dietary polyphenols and the age-related microglial phenotype via IL-8. Other evidence also corroborates the effect polyphenols have on reducing IL-8 activity in immune cells. Vitamin D_3_ decreased LPS-induced IL-8 production in human monocytes and macrophages in vitro [48], and EGCG significantly reduced IL-8 expression in macrophages after TNF-α stimulation [49]. Our findings merit further research on age-related IL-8 signaling in microglia and its attenuation by polyphenols given the magnitude of change between our *AGING* and *OLD-NT-020* comparisons.

Finally, we saw a reversal in predicted activity in the *Mechanistic Target of Rapamycin* (*mTOR*) *Signaling* pathway between *AGING* and *OLD-NT-020*. The mTOR pathway is a critical signaling junction that integrates multiple intra- and extracellular inputs to regulate microglial metabolism [50], growth and proliferation [51], and microglial activation [52], among other functions. We have also characterized mTOR as a prominent modulator of microglial dysfunction in aging mice [15] previously. In the present study, we show that mTOR signaling is predicted as activated in aging microglia (*z* = 1.342) but predicted as inhibited during aging with NT-020 supplementation (*z* = −2.449), implicating polyphenols as a possible mediator of the mTOR-driven aging microglial phenotype. Previous literature has identified how polyphenols affect cellular processes such as autophagy [53]. EGCG can rescue autophagy in hippocampal neurons in rats following chronic unpredictable mild stress [54]. This process was concomitant with improved cognitive function, decreased neuronal apoptosis, and reduced Aβ_1–42_ levels in the hippocampal CA1 region. Another study demonstrated that quercetin treatment restored locomotion in an Alzheimer’s disease model of *Caenorhabditis elegans* by activating proteasomal degradation of Aβ and inducing macroautophagy [55]. Regulation via mTOR is unique, since its activity is pleotropic in nature; therefore, a therapeutic such as NT-020 with effects as our data show represents a novel means of mitigating age-related microglial dysfunction.

### 2.4. NT-020 Diet Reverses Age-Related Changes in Predicted Rat Microglia Function

We used IPA to query functional pathways altered in aging rat microglia and examine how these changes manifested in old rats fed an NT-020-supplemented diet. We found that the following cell functions were predicted as activated based on the protein fold changes we observed in our *AGING* comparison: *activation of antigen presenting cells*, *synthesis of nitric oxide*, *aggregation of cells*, and *necroptosis* (Figure 4A).

Note that this network was generated using a significance cutoff at *p* < 0.10 to maximize protein count during pathway scoring, thereby improving the subsequent predictive analysis.

The age-related changes we found here support previous literature describing aging microglia. Increased antigen presentation was demonstrated in hippocampal microglia from 24-month-old mice relative to their 3-month-old counterparts [56]. These antigens included CD11b, and the microglial from the older mice were hyper-responsive to pro-inflammatory stimuli. Another study showed that greater numbers of inducible nitric oxide synthase (iNOS)-producing microglia have been found in aging mice in a Parkinson’s model where aggravation with MPTP amplified this effect [57]. Older microglia also exhibit increased aggregation in laser-induced brain injury models by way of lower process motility and migratory capacity [58]. Finally, necroptosis in microglia has been studied in retinal degenerative models where necroptotic microglia trigger neuroinflammation and neuronal cell death [59]. However, when we subsequently scored this functional pathway analysis with the *OLD-NT-020* dataset, we observed a reversal in the predictions reflected in the *AGING* comparison (Figure 4B). This inhibition is consistent with our observations in our differential expression and canonical pathway analyses, suggesting that NT-020 supplementation may reverse age-related changes in microglial function. Our findings are also consistent with other research regarding the polyphenols found in NT-020. EGCG reduced TLR4 signaling in dendritic cells after LPS stimulation via downregulated CD80, CD86, major histocompatibility complex I (MHCI), and MHCII [60]. EGCG treatment also decreased macrophage infiltration in the ankle joints of rats with college-induced arthritis [61]. Likewise, quercetin supplementation attenuated microglia-induced oligodendrocyte necroptosis via inhibition of Signal transducer and activator of transcription 1 (STAT1) and NF-κB pathways [62]. Finally, vitamin D_3_ decreased iNOS expression in macrophages cultured from human and rat kidneys via a STAT1–TREM-1 (Triggering receptor expressed on myeloid cells)-mediated process [63]. Our pathway analysis demonstrates the age-related molecular changes in microglia protein signaling and the reversal of those changes with polyphenol supplementation.

## 3. Conclusions

Aging is a complex and stochastic process at the cellular level. With age, cells lose their ability to maintain homeostatic mechanisms, and pathology can arise where certain cells become unable to perform basic functions that prevent organismal decline. Microglia bare a unique burden in the neuroimmune system in preserving an organism’s cognitive and motor function. Well-regulated microglia are critical for preventing age-related decline. Polyphenols represent a promising therapy in preserving microglial homeostasis in their potential to reverse the dysfunction microglia acquire during aging. We have identified a multitude of effects the polyphenol NT-020 exerts on the microglial proteome based on our results from MS-based proteomics and bioinformatics analyses. These effects result in a potential shift toward a more beneficial microglial phenotype during aging. However, we were unable to determine whether NT-020 supplementation affected microglia from younger animals. This result was likely due to the low number of differentially expressed proteins we detected in the *YNG-NT-020* comparison. Some studies have demonstrated a protective effect of polyphenols in younger animals [64,65] but the general trend in these studies points to a greater effect in older animals. A recent meta-analysis showed greater incidence of increased benefits from polyphenol supplementation in younger individuals [66]. However, this effect might be independent of an effect on neuroinflammation specifically. Improving MS techniques could increase the sensitivity and accuracy of protein detection in the future. This improvement would allow deeper insights regarding the interaction between polyphenols and aging microglial protein expression and resolve differences in younger and older animals. Under development are more optimized methods for quantitative profiling that enable the identification of over 6000 proteins via two-dimensional fractionation [67]. Such a technique reduces sample complexity and allows for a greater detection of low abundance proteins compared to the detection method used in our study. We were unable to quantify specific cytokines such as interleukin-1 (*IL-1*), interleukin-6 (*IL-6*), or tumor necrosis factor alpha (*TNF-α*) typically used to measure inflammatory states in microglia. Optimizing our detection protocols would provide enhanced proteome datasets to use in our downstream pathway analyses. Complex pathologies require comprehensive therapies for generating positive health outcomes and promoting well-being. Polyphenols offer such a means of combating aging pathology through the ease with which polyphenols are consumed through diet, their widespread availability, and their rising prevalence in the global market.

## 4. Materials and Methods

### 4.1. Animal Protocol and NT-020 Supplementation

All procedures were approved by the local institutional animal welfare committee. Male Fischer 344 rats either 3–5 months (YNG) or 20–22 months (OLD) of age were randomly segregated into two treatment groups: one group was fed an NIH31 control diet and the experimental group was fed a modified diet that included the NT-020 formulation at 135 mg/kg for 30 days (YNG-Tx & OLD-Tx).

NT-20 is a proprietary formulation that contains green tea extract which is a minimum of 95% polyphenols and 45% EGCG, blueberry powder from fruit, carnosine, 2000 IU Vitamin D3, and 40 mg grape seed extract. Initial studies describing the formulation described the process for choosing the formulation based upon screening many individual ingredients followed by targeted combinations; the choice of the final formulation showed more than additive effects of the four ingredients on stimulating the proliferation of stem cells in culture [26]. The dose of NT-020 used is based upon the recommended daily dose of NT-020 for humans. Dosing in animal models is adjusted for the metabolic rate, which is roughly 10× that of humans.

### 4.2. Microglial Extraction and Cell Culture

Rats were euthanized with CO2 according to IACUC standards. Brains were surgically removed following decapitation. Primary microglia were harvested as a single cell suspension with the MACS Neural Tissue Dissociation Kit (Miltenyi Biotec, San Jose, CA, USA, 130-092-628). The procedure is described briefly as follows: brains were placed in cold HBSS (w/o Ca+2, Mg+2) on ice. Brains were mechanically dissociated in a petri dish using a sterile scalpel and enzymatically digested with manufactured buffers; then, they were filtered through 70 µM cell strainers. An isotonic percoll solution was used to remove myelin from the cell suspension. After red blood cell lysis, microglia from cell suspensions were isolated via magnetic bead separation with microbeads specific to rat CD11b/c (Miltenyi Biotec, San Jose, CA, USA, 130-105-634). Total yield per brain was an estimated 1.5 million cells with 95% purity confirmed via immunostaining.

### 4.3. Sample Lysis, In-Gel Digest, and Label-Free Quantification

Primary microglia were lysed in a solution prepared from 4% SDS (*w*/*v*), 100 mM Tris-HCl (pH 7.6), 100 nM DTT, with protease and phosphatase inhibitors were added. The resulting lysate was sonicated at 20% amplitude in six-second pulses three times; then, it was centrifuged, and the supernatant was collected. Microglial protein extracts were separated by 1D SDS-PAGE followed by application of Coomasie stain for 30 min. Gels were de-stained with de-stain solution (50% methanol, 40% H_2_O, 10% acetic acid) twice in 15-min intervals; then, they were washed three times for 10 min using diH2O. Individual lanes were fractionated by excision of the lane into three separate gel regions; then, they were further cubed and placed in labeled 1.5 mL Eppendorf tubes. Samples were washed twice with acetonitrile (ACN) (50% in diH_2_O) and vortexed for 15 min. ACN was removed, and samples were rehydrated with 100 mM ammonium bicarbonate (ABC) solution for 5 min. An equivalent volume of ACN was added to samples followed by a 15-min vortex cycle, which was followed by a final aspiration and dry cycle in a speedvac for 5 min. Samples were rehydrated in 50 mM dithiothreitol (DTT) in 25 mM ABC solution and incubated for 30 min at 55 °C. When samples were returned to room temperature, DTT was removed, 100 mM iodoacetamide (IAA) in 25 mM ABC solution was added, and samples were incubated for 30 min. IAA was removed, and samples were washed three times for 15 min in 50:50 ACN/100 mM ABC. Buffer was removed, and samples were completely dried in a SpeedVac concentrator and then placed on ice for 5 min before being trypsinized overnight at 37 °C. Samples were desalted on C18 SPE columns, concentrated with a vacuum concentrator, and resuspended in 0.1% formic acid before MS analysis.

Relative protein quantification was performed by label-free MS-based quantitation. Digested peptides were separated on an Acclaim PepMap C18 (75 μm × 50 cm) UPLC column using an EASY-nLC 1000 with a gradient of 4–40% acetonitrile with 0.1% formic acid over 120 min. A top 10 data dependent acquisition (DDA) method with a scan range of 375–1500 *m*/*z* was utilized. The resolution of MS1 and MS2 were 70,000 (AGC of 1e6) and 17,500 (AGC of 2e5), respectively. Charge state exclusions included unassigned, 1, and greater than or equal to 8. Dynamic exclusion was set at 8 s. MS data were searched against the UniprotKB database for *Rattus norvegicus* (downloaded 02/26/2021 with 29,936 entries) using MaxQuant (ver. 1.6.17.0). Initial database search parameters included a precursor ion mass tolerance of 20 ppm with a recalibrated main search tolerance of 4.5 ppm, a minimum peptide length of 7 amino acids, and a maximum of 2 missed cleavages. Variable modifications included methionine oxidation and N-terminal protein acetylation. Cysteine carbamidomethylation was set as a fixed modification. LFQ intensities for each protein were first filtered for potential contaminants, only identified by site, and reverse sequences in excel before being analyzed in Perseus (ver. 1.6.1.1). Sample groups were then annotated, LFQ values were log_2_ transformed, and then, the groups were filtered for 75% valid values in total (among both replicates and comparisons). Missing LFQ values were imputed via Perseus with width and downshift parameters set to 0.4–0.7 and 1.4–1.6 in order to achieve appropriate lower end distribution of imputed values relative to the overall normally distributed log_2_ LFQ values for each group comparison [25] (see also Appendix A for full list of LFQ intensities across sample groups). Protein fold change ratios and Welch’s *t*-test differences were calculated from average LFQ intensities across the following comparisons: AGING (OLD/YNG), YNG-NT-020 ((YNG-Tx)/YNG), and OLD-NT-020 ((OLD-Tx)/OLD). Statistical significance was determined via Welch’s *t*-test in addition to a secondary filter of |z-score| > 1 to obtain appropriate FDR while maintaining sensitivity [25]. Fold change ratios and p-values for each comparison were uploaded into Ingenuity Pathway Analysis (IPA) for further predictive analyses.

### 4.4. Simple Western

Protein expression levels were measured via the WES instrument (ProteinSimple, San Jose, CA, USA, #004-600). Samples were prepared at 0.3 mg/mL in 0.1× sample buffer (ProteinSimple, 042-195) and combined with 5× Fluorescent Master Mix (ProteinSimple, PS-FL01-8) (1-part master mix to 4 parts lysate). Biotinylated ladder (MW 12-230 kDa) (ProteinSimple, PS-ST03EZ-8) was mixed with 16 µL deionized water, 2 µL 10× sample buffer, and 2 µL 400 mM DTT solution. Samples and ladder were denatured at 95 °C for 5 min, vortexed, and then placed on ice. CD11b (Novus, NB110-89474), Complement C3 (Novus, NBP1-32080), PGRMC1 (Novus, Centennial, CO, USA, NBP1-83220), Thioredoxin 1 (Cell Signaling, Danvers, MA, USA, 2429S), or GAPDH (Sigma-Aldrich, St. Louis, MO, USA, G9545) 1° antibodies were diluted separately in antibody diluent (ProteinSimple, 042-203). Chemiluminescent substrate (ProteinSimple, PS-CS01) was prepared by combining equal parts luminol-s and peroxide. Finally, ladder, samples, antibody diluent, primary antibodies, streptavidin-HRP (ProteinSimple, 042-414), anti-rabbit secondary (ProteinSimple, 042-206), and chemiluminescent substrate were pipetted on a WES microplate (ProteinSimple, SM-W003) with wash buffer (ProteinSimple, 042-202). Microplates were assayed under default protocol settings. Chemiluminescence in triplicate was obtained for each sample with each antibody, averaged, then normalized to average chemiluminescence for GAPDH.

### 4.5. Gene Ontology Analysis with GOplot in R

We performed gene ontology on differentially expressed proteins (*p* < 0.05) in the AGING and OLD-NT-020 comparisons using the GOplot package in R [26]. Circle plots generated with the GOCircle script integrated functional enrichment analysis on dataset proteins based on z-score using g: Profiler output [27].

### 4.6. Proteomic Comparisons in IPA

We used the IPA core analysis function to explore changes in canonical pathways and relevant cell function activity with age and NT-020 treatment. Molecules were analyzed for causal relationships against the Ingenuity Knowledge Base (reference set), and both direct and indirect relationships were considered. Reference data in IPA were sourced from Ingenuity Expert Findings and Ingenuity Supported Third Party Information. We adjusted the p-value cutoff for analysis to *p* < 0.1 to increase the number of functional associations between dataset proteins. Canonical pathways were scored for p-value of overlap (p(r), which the measures enrichment of experimental proteins relative to the Ingenuity Knowledge Base and activation z-score, which infers predicted regulation relative to downstream protein expression. Predicted activity in annotated cell functions was determined via z-score.

## Figures and Tables

**Figure 1 ijms-22-06373-f001:**
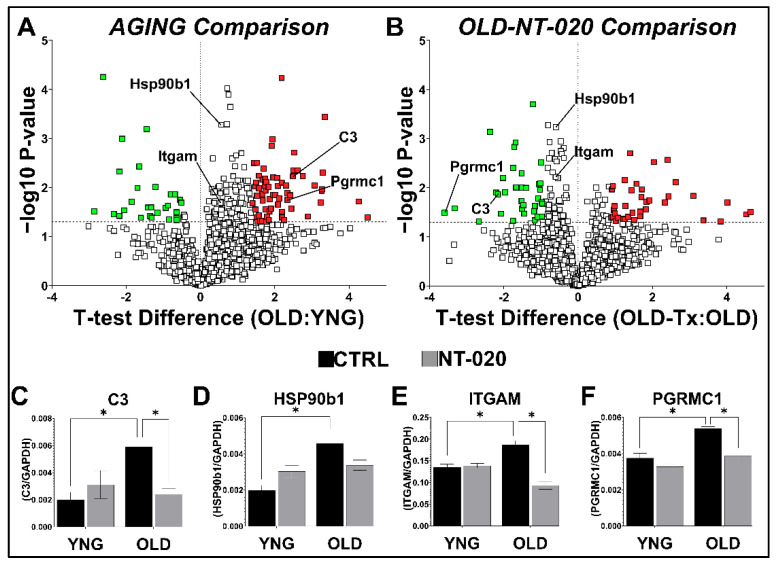
Differential microglial protein expression in *AGING* and *NT-020* comparisons. Protein fold change was calculated using average LFQ intensities among samples (*n* = 4 rats per group) across three comparisons: *AGING* ($\frac{OLD}{YNG}$), *YNG-NT-020* ($\frac{YNG-Tx}{YNG}$), and *OLD-NT-020* ($\frac{OLD-Tx}{OLD}$). Welch’s *T*-test difference values were plotted against the corresponding −log_10_(*p*-value) for each protein to generate volcano plots; the horizontal dotted line indicates the cutoff for statistical significance via Welch’s *t*-test (*p* < 0.05); green (downregulated) and red (upregulated) color indicate proteins with z-score > 1 (**A**,**B**). Protein expression for inflammatory mediators such as *ITGAM*, *C3*, *HSP90B1*, and *PGRMC1* followed the trend of being upregulated with age and downregulated with age and the NT-020 diet. MS results were validated using the Wes instrument (**C**–**F**). The average chemiluminescence ± SEM was obtained for 1° antibodies against *ITGAM*, *C3*, *HSP90B1*, and *PGRMC1* (normalized to *GAPDH*) across treatment groups (*n* = 5 rats per group) and two-way ANOVA analysis was performed with Tukey’s post-hoc test (*, *p* < 0.05).

**Figure 2 ijms-22-06373-f002:**
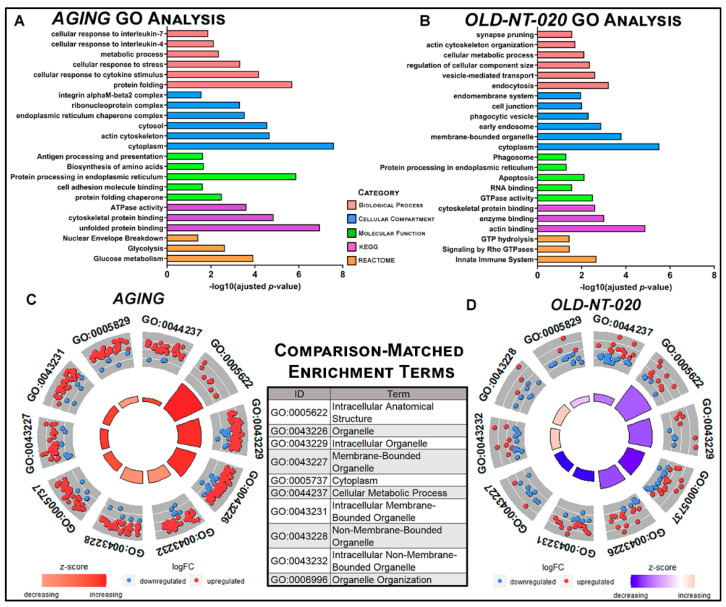
Gene ontology analysis in differentially expressed microglial proteins in *AGING* and *OLD-NT-020* comparisons. Representative GO results queried in g: Profiler for *AGING* (**A**) and *OLD-NT-020* (**B**) comparisons. Terms on vertical axis faceted among biological process, molecular function, cellular compartment and annotations in KEGG and REACTOME pathways. Significance of terms is plotted on the horizontal axis using −log_10_(adjusted *p*-value). Circle plots generated with the GOCircle function for top 10 enrichment terms by z-score shared by *AGING* (**C**) and *OLD-NT-020* (**D**) comparisons. Outer circle scatterplot represents log_2_(fold-change) of proteins annotated to terms; red circles = upregulation, blue circles = downregulation. Inner bars depict z-score (calculated as $\frac{\left(\#\;of\;upregualted\;proteins-\#\;of\;downregulated\;proteins\right)}{\sqrt{\#\;of\;proteins\;per\;GO\;term}}$) where red = increasing z-score and blue = decreasing z-score. *AGING* showed marked enrichment for upregulated proteins within terms while *OLD-NT-020* trended oppositely.

**Figure 3 ijms-22-06373-f003:**
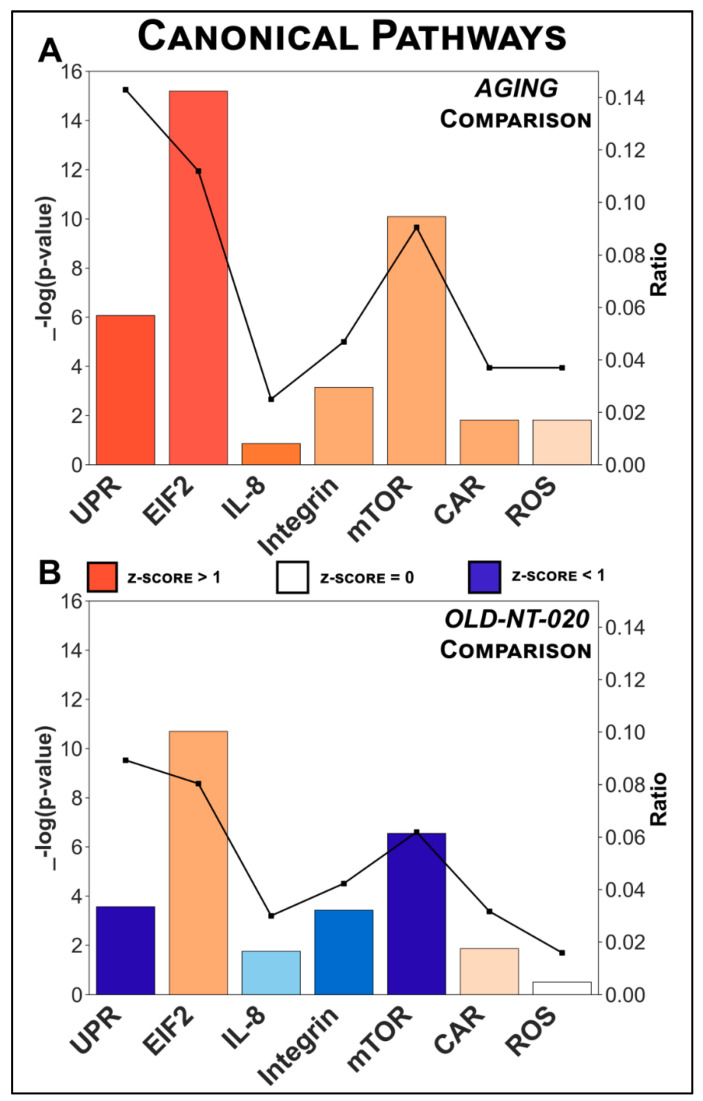
Canonical pathways in rat microglial proteome altered by age and diet. Canonical microglial signaling pathways annotated by IPA’s core analysis module for *AGING* (**A**) and *OLD-NT-020* (**B**) comparisons. *X*-axis indicates pathways (*UPR* = unfolded protein response, *EIF2* = eukaryotic initiation factor 2 signaling, *IL-8* = interleukin-8 signaling, *Integrin* = integrin signaling, *mTOR* = mechanistic target of rapamycin signaling, *CAR* = xenobiotic metabolism CAR signaling pathway, *ROS* production = production of nitric oxide and reactive oxygen species in macrophages), left *Y*-axis denotes −log(*p*-value) derived from Fischer’s exact test right-tailed, right *Y*-axis (points on black line) represents the ratio of dataset proteins to total known proteins for that pathway. Orange bars indicate predicted activation (positive z-score), while blue bars indicate predicted inhibition (negative z-score). A general trend is observed where pathways in *AGING* have reverse predicted regulation in *OLD-NT-020*. *UPR*, *IL-8*, *Integrin*, and *mTOR* pathways exhibited the greatest absolute change in activation z-scores across comparisons.

**Figure 4 ijms-22-06373-f004:**
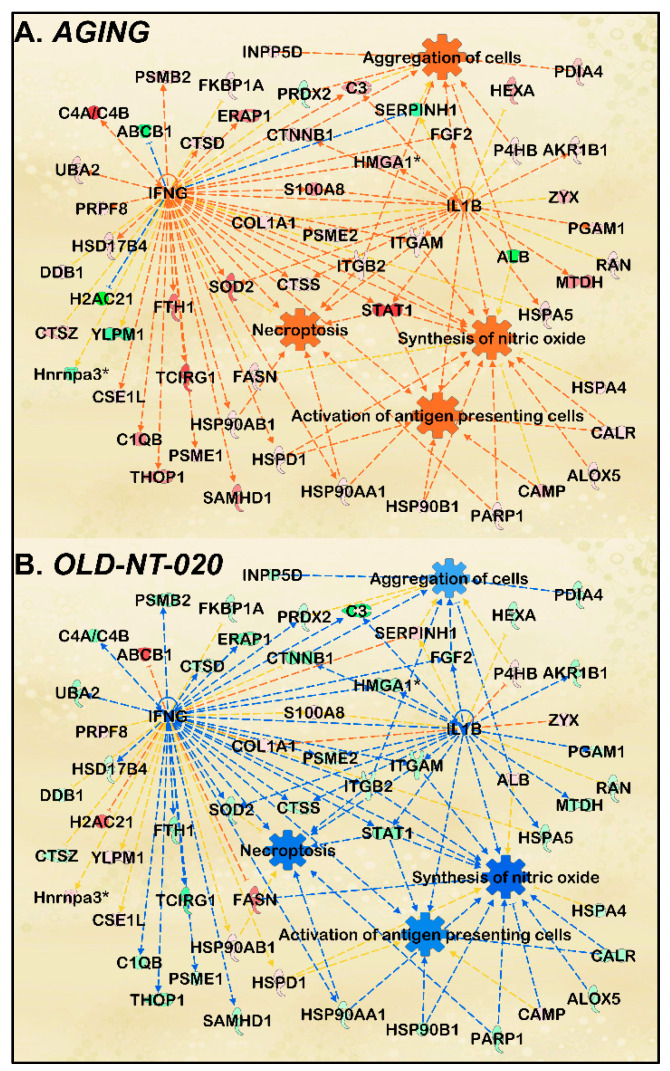
Microglial functions modified during aging and with polyphenol supplementation. IPA network analysis featuring annotated microglial functions enriched by MS-detected proteins for *AGING* comparison (**A**). Orange and blue nodes represent predicted activated and inhibited regulators, respectively. Darker colors indicate greater predicted activation or inhibition. Orange and blue lines depict regulation from upstream molecules to downstream function. Dataset proteins shown in green (downregulated) or red (upregulated) with MS-determined LFQ fold change and *p*-values; darker colors represent greater fold change; significance cutoff at *p* < 0.1 to incorporate a greater number of dataset proteins in predictive analysis. Annotated functions included antigen presentation, nitric oxide synthesis, cell aggregation, and necroptosis. Then, the pathway was fitted with *OLD-NT-020* dataset proteins (**B**) to observe changes in aging microglial cell functions with NT-020 diet. Consistent with trends observed in differential protein expression, gene ontology, and canonical pathway analysis, activated predictions in *AGING* were reversed to inhibition in *OLD-NT-020*.

## Data Availability

The data presented in this study are available in Appendix A: Dataset Proteins.

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
