# Peer review of "Polyphenol Supplementation Reverses Age-Related Changes in Microglial Signaling Cascades"

_ijms, 2021, doi:10.3390/ijms22126373_

Round 1

Reviewer 1 Report

The Authors in the present original article evaluated, using a MS-based proteomics and related bioinformatics analyses, the effects that NT-020 (a proprietary blend of blueberry, green tea, vitamin D3, and l-carnosine) exerts on the microglia. In particular, They underlined the fundamental involvment of microglia in the pathophysiological process(es) of aging and observed that NT-020 may reverse age-related canonical pathways.

 Good article. I think that the present manuscript has an interesting and actual topic and it could be a starting point for a better understanding of the pathophysiological process(es) involved in CNS aging and, interestingly, it provided important evidence that NT-020 treatment is able to rescue microglial age-related alterations.

In my opinion, this manuscript has a clear message, the rationale for the choice of the experimental model as well as the technical approaches used are appropriate. The obtained results are fully described. However, in my opinion, the Authors have to better describe and discuss the formulation used, also due to it is not only characterized by polyphenols. It is important to describe better NT-020 and also discuss the potential involvement of the other biomolecules and the probable synergic effects of the NT-020 components.

Furthermore, I have other mandatory comments:

- In all the text check that abbreviations are reported in full the first time that are cited.

Introduction

- Some references that justify the sentences reported are missing;

- briefly amply the polyphenols description and their involvent in aging.

Materials and Methods

- Add, if possible, one or more references that justify the animal model and NT-020 dose, time and formulation of treatment used (or add a brief personal explanation about it).

Discussion

- Discuss the formulation used underling the features of the components and their involvement in aging;

- A simple schematic graph that summarized the data obtain should be useful for the readers.

Author Response

Thank you for the comments that this is a good and interesting article with a clear message.

Revisions:

Discuss the formulation:  We have added a full description of NT-020 in the methods section (see Materials and Methods 4.1):

“NT-020 is a proprietary formulation that contains green tea extract which is a minimum of 95% polyphenols and 45% EGCG, blueberry powder from fruit, carnosine, and 2000 IU Vitamin D3, and 40 mg grape seed extract.   Initial studies describing the formulation described the process for choosing the formulation from screening many individual ingredients and followed by combinations, the choice of the final formulation showed more than additive effects of the four ingredients on stimulating proliferation of stem cells in culture. The dose of NT-020 used is based upon the recommended daily dose of NT-020 for humans. Dosing in animal models is adjusted for metabolic rate which is roughly 10X that of humans.”   

Introduction: We have added additional references. We have also added more information about polyphenols and their involvement in aging (see Introduction, paragraph 3), and included more information on the formulation used (see Introduction, paragraph 4).

Methods: as discussed above we have added a description of the formulation to the methods section.

Discussion: A graphical abstract has been added in response to the request for a schematic summarizing the data.

English language comment: A spell check has been performed and typographical and grammatical errors corrected.

Reviewer 2 Report

The present manuscrit entitled "Polyphenol supplementation reverses age-related changes in microglial signaling cascades" report the results obatined after comparing protein levels found between primary microglia isolated from young and aged rats a and identifying canonical signaling cascades related to microglial homeostasis and function affected by age. Likewise, it has been evaluated the effect of supplementing with a blend of polyphenols animals of both ages in these pahtways and protein levels. The findings of this study are very interesting taken into account the relevance of microglial age-associated dysfunction for nervous system aging and its associated diseases. In general, conclusions and discussion are supported by the results, but  some issues could be imporved before publication.

Major revisions:

-It its possible to know more details about the supplement used? e.g. total phenol content. Are there some special storage conditions for diet withthe NT-020 formulation?

Minor revisions:

-In material and methods section, please improve the references for the reagents and material used.

-Authors report and discussed interesting effects of the supplement on mTOR pathway, but its role in autophagy and proteostasis maintenance should be mentioned. In this sense, many bioactive compounds including phenolics have shown interesting effects (see doi: 10.1089/ars.2017.7234)

- Has the NT-020 supplementationeffect on young animals respect than young controls? Some effects in that sense also could be relevant to prevent age-associated alterations if the indivuals start to take supplements at earlier ages.

Author Response

We thank the reviewer for the positive comments on our manuscript.

Major Revisions:

We have added a description of the formulation and how NT-020 was designed to the manuscript (see Materials and Methods 4.1 and Introduction, paragraph 4).

Minor Revisions:

We have added information on the reagents used into the methods section (see Materials and Methods 4.4.).

The role of autophagy has now been added to the discussion and the paper mentioned cited (see Introduction, paragraph 3).

The NT-020 supplementation was given to young subjects as well. We did not discuss this data in detail in the initial submission as there were no large changes in the expression of microglial proteins. We observed only 36 differentially expressed proteins in the young subjects compared with young given NT-020 which was not sufficient to perform meaningful bioinformatic analysis. We have added additional discussion of this in the revised manuscript (see Conclusions).

Round 2

Reviewer 2 Report

The manuscript can be published in the present form